# WWOX Loss of Function in Neurodevelopmental and Neurodegenerative Disorders

**DOI:** 10.3390/ijms21238922

**Published:** 2020-11-24

**Authors:** C. Marcelo Aldaz, Tabish Hussain

**Affiliations:** Department of Epigenetics and Molecular Carcinogenesis, Science Park, The University of Texas MD Anderson Cancer Center, Smithville, TX 78957, USA; ftabish@mdanderson.org

**Keywords:** WWOX, spinocerebellar ataxia, epileptic encephalopathy, WOREE, intellectual disability, autism, ADHD, multiple sclerosis, Alzheimer’s disease, neurodegeneration

## Abstract

The *WWOX* gene was initially discovered as a putative tumor suppressor. More recently, its association with multiple central nervous system (CNS) pathologies has been recognized. *WWOX* biallelic germline pathogenic variants have been implicated in spinocerebellar ataxia type 12 (SCAR12; MIM:614322) and in early infantile epileptic encephalopathy (EIEE28; MIM:616211). *WWOX* germline copy number variants have also been associated with autism spectrum disorder (ASD). All identified germline genomic variants lead to partial or complete loss of WWOX function. Importantly, large-scale genome-wide association studies have also identified *WWOX* as a risk gene for common neurodegenerative conditions such as Alzheimer’s disease (AD) and multiple sclerosis (MS). Thus, the spectrum of CNS disorders associated with WWOX is broad and heterogeneous, and there is little understanding of potential mechanisms at play. Exploration of gene expression databases indicates that *WWOX* expression is comparatively higher in the human cerebellar cortex than in other CNS structures. However, RNA in-situ hybridization data from the Allen Mouse Brain Atlas show that specific regions of the basolateral amygdala (BLA), the medial entorhinal cortex (EC), and deep layers of the isocortex can be singled out as brain regions with specific higher levels of *Wwox* expression. These observations are in close agreement with single-cell RNA-seq data which indicate that neurons from the medial entorhinal cortex, Layer 5 from the frontal cortex as well as GABAergic basket cells and granule cells from cerebellar cortex are the specific neuronal subtypes that display the highest *Wwox* expression levels. Importantly, the brain regions and cell types in which WWOX is most abundantly expressed, such as the EC and BLA, are intimately linked to pathologies and syndromic conditions in turn associated with this gene, such as epilepsy, intellectual disability, ASD, and AD. Higher *Wwox* expression in interneurons and granule cells from cerebellum points to a direct link to the described cerebellar ataxia in cases of WWOX loss of function. We now know that total or partial impairment of WWOX function results in a wide and heterogeneous variety of neurodegenerative conditions for which the specific molecular mechanisms remain to be deciphered. Nevertheless, these observations indicate an important functional role for WWOX in normal development and function of the CNS. Evidence also indicates that disruption of WWOX expression at the gene or protein level in CNS has significant deleterious consequences.

## 1. Introduction

The WW domain-containing oxidoreductase gene (*WWOX*), originally discovered by our laboratory, maps to the ch16q23.1-23.2 region and encodes a 414-amino acid protein composed of two WW domains in its N-terminus and a central short-chain dehydrogenase/reductase (SDR) domain [1]. The *WWOX* genomic locus contains a hotspot for genomic instability which is *FRA16D*, the second most common chromosomal fragile site in the human genome, susceptible to the occurrence of breaks and rearrangements commonly causing germline and somatic copy number variants (CNVs) [1,2]. Somatic deletions and translocations affecting *WWOX* accompanied by loss of expression are frequent in multiple cancer types and associated with tumor progression, therapy resistance, and poor disease outcomes [3,4]. *WWOX* was initially described by us and others as a putative tumor suppressor gene. However, since biallelic or monoallelic (i.e., haploinsufficiency) *WWOX* deletions are not tumorigenic in most animal models or humans, it can be safely concluded that *WWOX* does not behave as a classical or highly penetrant tumor suppressor but rather as a facilitator or co-driver of tumor progression when lost [3].

In recent years, abundant evidence from multiple studies has accumulated causally linking WWOX loss of function with various central nervous system (CNS) pathologies. As it will be discussed in the following sections, human *WWOX* germline pathogenic variants have been directly implicated in complex and heterogeneous neurological disorders [5,6,7,8,9,10,11]. Similarly, mouse and rat *Wwox* null models, either genetically engineered or spontaneously occurring, also display phenotypes with various neurological abnormalities [5,12,13]. Thus, findings from diverse species point to a significant role for WWOX in the normal development and function of the CNS. Here, we discuss key evidence supporting the aforementioned conclusions while summarizing information from various CNS expression datasets and discussing cellular pathways in which WWOX is known to play relevant roles that are potentially associated with the various neurodegenerative conditions.

## 2. WWOX Expression in CNS

WWOX mRNA and protein are known to be ubiquitously expressed in multiple tissues displaying the highest levels in thyroid, cerebellum, brain, and reproductive organs [3,14]. Early mouse studies described Wwox expression in derivatives of ectodermal layers during embryonic development, including cells from CNS, peripheral nervous system (PNS), and sense organs. Wwox expression was described as relatively low in the newly formed neural tube and primary brain vesicles at E8–E9 and increasing to high levels, particularly in the brainstem, spinal cord, and peripheral nerve bundles from E12 onward, followed by a significant decrease after birth. In late developmental stages, neural crest-derived structures, including cranial, spinal ganglia, and cranial mesenchyme, showed the highest levels of Wwox expression. In the adult mouse brain, Wwox was described as abundantly expressed in the epithelial cell layer of the choroid plexus and ependymal cells while low to moderately expressed in the cerebral cortex, striatum, optic tract, and cerebral peduncle [15]. More recently, in rats, Wwox was found expressed in almost all brain regions, including the olfactory bulb, cerebral cortex, hippocampus, diencephalon, cerebellum, brain stem, spinal cord, and in various cell types including neurons, astrocytes, and oligodendrocytes [16].

In early observations, we showed robust WWOX protein expression in soma from cells of all three layers of human cerebellar cortex viz., Purkinje cell layer, molecular layer, and granular layer. WWOX-positive immunostaining was also observed in cerebrum samples, including soma and dendrites of pyramidal neurons from frontal and occipital cortices and nucleus caudate, in pons and nuclei olivaris of the medulla, and astrocytes from all such regions. Neuropils and small neurons were also immunoreactive to the WWOX antibody, while low to negative expression was observed in substantia nigra. Additionally, neurons at autonomic ganglia also showed intense cytoplasmic WWOX staining [14].

In order to evaluate comparative temporospatial expression of *WWOX* in human CNS, we explored the Human Brain Transcriptome (HBT) dataset (https://hbatlas.org/), which is based on Affymetrix GeneChip arrays [17]. As it can be observed in Figure 1a, *WWOX* mRNA expression is quite uniform from conception to adulthood in all depicted brain regions, including: neocortex (NCX), hippocampus (HIP), amygdala region (AMY), striatum (STR), mediodorsal nucleus of the thalamus (MD), and cerebellar cortex (CBC). All regions show relatively higher expression levels in early embryonal life slowly decreasing during fetal development until birth to slightly increase again *WWOX* expression in postnatal life and early childhood up to adolescence (periods 8 through 12 in Figure 1a), remaining quite stable thereafter. It must be noted however, that *WWOX* expression in cerebellar cortex (red line) clearly behaves differently showing a more significant increase in early postnatal life and remaining higher (compared to other tissues) up to adolescence (period 12 in Figure 1a). By exploring the RNA-seq GTEx database (https://gtexportal.org) [18], indeed, it can be observed that cerebellum is the CNS structure with the highest *WWOX* expression levels in adults (median = 12.1 TPM) and the expression of this gene is observed throughout the various CNS tissues with the frontal cortex, amygdala, and hippocampus expressing 57–46% of the levels seen in the cerebellum (median for FC = 6.9, AMY = 6.2, HIP = 5.6 TPM, Figure 1b).

Interestingly, however, in-situ hybridization (ISH) data of mouse brain from the Allen Brain Atlas (https://mouse.brain-map.org/) [19] show that *Wwox* ISH probes significantly light up a region of hippocampal formation, specifically layer 2 of the medial entorhinal cortex (ENTm2, medial part, dorsal zone, layer 2) and also the anterior (BLAa) and posterior (BLAp) regions of the basolateral amigdalar nucleus in the cortical subplate (Figure 2a,b). *Wwox* probe hybridization can also be detected, to a much lesser extent, in the somatosensory and somatomotor Layer 5 areas of isocortex, and is also clearly visible in cerebellum cortex (Figure 2b). Present at the interface of neocortex and hippocampus, the entorhinal cortex (EC) acts as the nodal point which receives highly processed inputs from every sensory modality relating to ongoing cognitive processes, which in turn are conveyed to the hippocampal formation. The EC-hippocampus system thus regulates important brain functions and acts as a hub of networks mediating learning, memory, navigation, and sense of time [20]. Learning and memory are in turn specifically influenced by emotions which are processed by the amygdala. The basolateral amygdala (BLA) is one of the three regions of amygdalar nuclei which plays an integral role on emotional behavior, particularly fear, stress, and anxiety. Some studies suggest that BLA and hippocampus function synergistically, where one processes emotions and the other stores memories, respectively [21].

Wwox cell-type-specific expression profiles were explored in a publicly available mouse brain RNA-seq database (www.BrainRNAseq.org) [22,23]. At P7, uniform expression of *Wwox* is detected in neurons and all glial cell types (Figure 3a,b). Within the oligodendrocyte population, the highest *Wwox* expression levels are observed in progenitor oligodendrocytes, with significantly lower levels in mature myelinated oligodendrocytes (Figure 3b). This suggests that Wwox might play a role during the development and differentiation of these specialized cells. *Wwox* expression remains stable throughout life in microglial cells, the resident immune cells of the CNS (Figure 3c). Interestingly, substantial upregulation of *Wwox* expression can be observed in microglial cells upon treatment of mice with lipopolysaccharide (LPS) used as an inflammatogenic agent (Figure 3d). This suggests that WWOX might play a role in CNS inflammatory response. In agreement with this observation, evidence from studies in various inflammatory pathologies strongly suggests that Wwox has a relevant function in inflammation in general. Furthermore, a direct association between WWOX and the NF-kB inflammation signaling pathway has been demonstrated [24,25,26,27].

In order to analyze *Wwox* expression at the single cell level, we explored the mouse brain single-cell RNA sequencing (scRNA-seq) DropViz database (http://dropviz.org/) [28]. Specific neuronal and interneuronal subtypes from cerebellum, frontal cortex, and the hippocampal region show distinctively higher levels of *Wwox* expression relatively to all other neurons in the CNS and the results are in close agreement with the described observations from GTEx and the Allen Mouse Brain Atlas.

The neuronal cell clusters that express the most *Wwox* transcripts include interneurons (Pvalb+ cells) and granule cells (Gabra6+) from the cerebellum (Figure 4a). In Figure 4b, these two clusters are depicted in a tSNE plot as the darkest colored regions among cerebellum cells. By dissecting the specific subclusters expressing the most *Wwox* transcripts in the cerebellum, it can be observed that specifically GABAergic basket cells (i.e., interneurons) and, to lesser extent, granular neurons are the top cell types (Figure 4c). This is in agreement with the above described observations of the cerebellum being a key region of CNS expressing higher WWOX levels in humans and mice and provides an insight into the specific more relevant neuronal types within this structure.

Markers Parm1, Syt6, and Fezf2 identify the clusters of neurons expressing the most *Wwox* in frontal cortex (Figure 5a) and these three clusters are singled out in the tSNE plot of Figure 5b. Upon further subclustering, it can be clearly observed that deep-layer pyramidal neurons, specifically from Layer 5, show the highest levels of *Wwox* expression (Figure 5c). This is also in agreement with observations from the Allen Atlas, pointing to Layer 5 areas of isocortex as a region of increased *Wwox* expression by ISH.

Finally, markers Lhx1, Nxph3, and Gad2 identify the neuronal clusters expressing the most *Wwox* in hippocampus as also noted in the tSNE plot (Figure 6a,b). Further dissection by subclustering highlights very specifically neurons from the medial entorhinal cortex (identified by Slc17a7 and Reln markers) as those cells expressing, by far, the highest *Wwox* number of transcripts in hippocampal formation (Figure 6c). This is in striking close agreement with the observations described in Figure 2b which specifically shows significant detected *Wwox* expression by ISH in layer 2 of the medial entorhinal cortex, dorsal part (ENTm2 region). Other non-neuronal cell types that at the single cell level display significant number of *Wwox* transcripts in the mouse CNS include ependymal cells and choroid plexus cells (data not shown).

In summary, the cerebellar cortex is the CNS site with highest WWOX expression in humans and rodents. From exploring scRNA-Seq mouse datasets, GABAergic basket cells and granular cells appear as the subtypes with highest expression levels. Specific regions of the medial entorhinal cortex and the basolateral amygdala show focalized *Wwox* expression, followed by deep layers of the frontal cortex, in particular layer 5.

## 3. WWOX-Associated CNS Disorders

### 3.1. WWOX in Spinocerebellar Ataxia Type 12 (SCAR12)

The first human neuropathology associated with homozygous missense *WWOX* mutations was SCAR12 (MIM: 614322). This condition was first described by Gribba et al. as a novel early-childhood onset recessive cerebellar ataxia associated with generalized tonic-clonic epilepsy and severe intellectual disability in four siblings from a large consanguineous family from Saudi Arabia. The onset of seizures was recorded at a very early age, between 9 and 12 months in all four siblings. Moderate to mild cerebellar ataxia and psychomotor retardation was noticed when the children started to walk, which was delayed until the age of 2–3 years. All four patients suffered from severe dysarthria, nystagmus, diminished reflexes, and severe intellectual disability. Mild cerebellar atrophy was seen in brain magnetic resonance imaging (MRI) of two affected children. In that first study, using homozygosity mapping, the defective gene was mapped to a 19-Mb region at ch16q21-q23 [29]. In 2014, by means of whole-exome sequencing, it was determined that the four affected children harbored a homozygous missense mutation of the *WWOX* gene. The novel p. Pro47Thr mutation affects a highly conserved proline residue and critical component for the function of the first WW domain of the protein (Figure 7a and Appendix A). We demonstrated that this mutation renders WWOX unable to bind canonical PPxY or similar motifs in interacting partner proteins [5]. In the same study, a second consanguineous Israeli-Palestinian family with two affected children was described. It was observed that both patients carried a homozygous mutation at another highly conserved WWOX protein residue, pGly372Arg (Figure 7a and Appendix A). The phenotype of both affected siblings in that second family resembled the index family with disease onset in the first two years of life with generalized tonic-clonic epilepsy, developmental delay, intellectual disability, spastic ataxia, and paraplegia [5]. The pGly372Arg mutation affects a glycine at the C-terminus in the short-chain dehydrogenases/reductases (*SDR*) domain of WWOX (Figure 7a) and its functional consequence is unclear at this point. Analysis from patient fibroblasts showed that both described missense mutations do not alter WWOX protein expression and instead result in the production of a defective, partially functional WWOX protein (i.e., hypomorphic mutations) [5].

### 3.2. WWOX-Related Early Infantile Epileptic Encephalopathy (WOREE)

Early infantile epileptic encephalopathies (EIEE) are a group of early-age brain disorders characterized by unremitting seizures and epileptic activity, which have a long-lasting deleterious impact on the developing brain and contribute to progressive cognitive and neurological deterioration. In recent years, numerous genes have been identified to be causally associated with various EIEE and *WWOX* has recently been recognized as one of them [30]. Soon after the first *WWOX* familial mutations were linked to SCAR12, new familial mutations, such as homozygous missense, nonsense, splice-site mutations, or compound heterozygous mutations, combining as well with CNVs and ultimately leading to complete loss of *WWOX* function, were associated with early infantile epileptic encephalopathy (EIEE28; MIM: 616211), a condition also described as *WWOX*-related epileptic encephalopathy (WOREE) syndrome [7,8,9,31]. The most distinguishing feature of these patients includes pharmaco-resistant epilepsy starting as early as the first week of life. The affected children usually show profound developmental abnormalities, spastic quadriplegia of early-onset, psychomotor delay, absence of language development, failed acquisition of walking, and commonly, death within the first two years of life. Many cases also display microcephaly and ophthalmic abnormalities, such as poor or no eye contact, retinal degeneration, and optic atrophy. Brain MRI of these patients usually shows reduced myelination, brain structural defects of varying degrees, such as loss of volume in supratentorial structures including thinning of the corpus callosum, progressive cerebral atrophy involving grey and white matter, particularly in the frontal and temporal regions, as well as widening of Sylvian fissures. Reviews of the specific clinical and molecular features of WOREE cases have been published [7,8,9]. We summarize clinical and mutational information from these studies, along with additional cases described subsequently, in Appendix A. In Figure 7a, we depict the specific missense, nonsense, and splice-site variants affecting the WWOX protein in reported WOREE cases. Germline CNVs affecting the *WWOX* locus in WOREE cases are shown in Figure 7b, where deletions are shown in black, and duplications in red. All CNVs, except one, are either intragenic or have breakpoints within the genomic region spanned by WWOX, and deletions are predominant compared to duplications.

There is a considerably uniform distribution of point mutations throughout the WWOX protein; however, a few hotspots for the occurrence of more than one mutation at the same amino acid site, or mutation affecting different families, can be observed (indicated in red lettering in Figure 7a). For instance, the p.Gln230Pro mutation was found both as a biallelic mutation and also in heterozygous compound cases combined with other mutations and has been reported in a total of 8 WOREE cases affecting a total of 6 different families. This mutation is found within the large SDR domain of WWOX, not obviously affecting a critical enzymatic functional motif. However, two other mutations affect Glycine 137 which is a key residue of the coenzyme binding region have been reported (Figure 7a). In addition, compound heterozygous mutations affecting glutamic acid 17 and serine 318 combined with deletion or other missense mutations have also been identified. Another possible mutational hotspot is indeed the extremely conserved residue proline 47 in the first WW domain of WWOX and which is biallelically mutated to threonine (p.Pro47Thr) in SCAR12 [5] and, in two reported WOREE cases, the mutation p.Pro47Arg affecting one allele is combined with a frameshift mutation in the remaining allele (Figure 7a and cases ID 12 and 13 in Appendix A). Interestingly, although WWOX is predominantly a cytoplasmic protein, prediction of a high confidence putative nuclear localization signal (NLS) starts precisely at proline 47 (PKTGKRKRVAG) as per cNLS Mapper (http://nls-mapper.iab.keio.ac.jp/cgi-bin/NLS_Mapper_help.cgi) [32]. This suggests that mutation at proline 47 can not only affect WWOX interaction with partner proteins, but it can also affect translocation to the nuclear compartment. Currently, the mechanistic effects of most of the observed missense mutations on WWOX function are not known, therefore, it would be of interest to undertake studies exploring the functional impact of specific hotspot mutations since could lead to identify additional protein functional motifs and in turn provide further mechanistic insight. There are certainly also some hotspots for recurrent non-sense mutations, e.g., Arg54*, frameshift mutations e.g., Asp58Alafs*3, His173Alafs*67, His173Ilefs*5, Glu306Aspfs*21, as well as splice site mutations e.g., c.173-1G>T, c.173-2A>G, c.409+1G>T, c.517-2A>G, and c.606-1G>A (Figure 7a). In addition, recurrent deletions and duplications of exons 6-8 are also reported in many cases from different families (Figure 7b). These mutations lead to truncated or scrambled protein products which likely rapidly undergo degradation; however, the explanation for this type of hotspot should likely be searched at the genomic sequence level.

### 3.3. Evidence from Rodent Models of Wwox Ablation

The neurodevelopmental deficits observed in humans due to WWOX function impairment are recapitulated in Wwox loss of function rodent models. The first CNS related observations were described in a spontaneously mutated rat strain displaying lethal dwarfism and epilepsy (*lde*). In later studies, the *lde* locus was mapped to rat chromosome 19 and Wwox was identified as the culprit gene. Specifically, *lde/lde* rats carry a homozygous 13-bp deletion affecting exon 9 of Wwox. The *lde/lde* rats have a short lifespan reaching only 3–12 weeks of age, displaying growth retardation, high incidence of spontaneous and audiogenic induced epileptic seizures, ataxic gait, and motor dysfunctions [12,13]. In contemporaneous studies, while developing models to study the role of WWOX in cancer, we generated a full *Wwox* knockout (*Wwox-KO*) model using mice harboring loxP sites flanking exon 1 of the gene (*Wwox^loxP/loxP^*) crossed to mice expressing Cre-recombinase controlled by adequate promoters to obtain whole-body deletion [33]. *Wwox-KO* mice display significant growth retardation (dwarfism) and metabolic deficits, including severe hypoglycemia, hypocalcemia, as well as signs of metabolic acidosis and kidney failure [33]. These mice die postnatally as early as 72 h after birth, with none living more than 3–4 weeks, a phenotype in terms of early death similar to the *WWOX* null genotype in WOREE children [33]. In the study of Mallaret et al. [5], with the first description of familial cases with WWOX germline mutations, we described that *Wwox-KO* mice display spontaneous and audiogenic tonic-clonic seizures at a very early age as well as ataxic gait and decreased mobility, matching observations in *lde/lde* rats [3,5]. Notably, heterozygous *Wwox-KO/WT* mice, similar to mutation carrier parents of WOREE children, failed to display any distinct abnormal CNS phenotypes or significantly decreased overall survival.

In follow-up studies, we observed that *Wwox* deletion led to a significant reduction in the number of parvalbumin and neuropeptide Y expressing GABAergic interneurons in the hippocampus of *Wwox-KO* mice [34]. Interneurons use the inhibitory neurotransmitter GABA and coordinate neuronal activity by regulating the flow of signals and network synchronization. Genetic mutations resulting in selective loss or functional impairment of GABAergic interneurons are known to disrupt the regulation of local excitatory and inhibitory circuits, causing hyperexcitability of neuronal networks, thus contributing to epileptogenesis [35,36]. We also observed reduced expression of glutamic acid decarboxylase proteins GAD65/67 in *Wwox* null hippocampi; these are the principal enzymes responsible for GABA synthesis, overall suggesting lower levels of GABA in brains of KO mice [34]. The decrease abundance of GABAergic interneurons and reduced expression of GAD65/67 in *Wwox* null mice suggests the existence of a hippocampal hyper-excitable state, explaining, at least in part, the epileptic phenotype observed in these mice [34].

Comparative transcriptome analyses of neural stem cells (neurospheres) generated from *Wwox-KO* vs. *WT* hippocampi, identified enrichment of biofunctions related to neurological diseases and CNS development. Several epilepsy-related genes were found differentially expressed in *Wwox-KO* neurospheres. Importantly, we also observed that Wwox deficiency was associated with increased neuroinflammation since *Wwox-KO* hippocampi displayed increase in activated microglia and clear evidence of astrogliosis as well as overexpression of inflammatory cytokines such as TNF-a and IL6 [34]. Clinical and experimental observations have provided strong evidence that inflammation may constitute a crucial mechanism in the pathophysiology of CNS disorders [37]. The finding of elevated inflammation in *Wwox-KO* mice brains in addition to the previous described observation of increased *Wwox* expression in microglial cells upon exposure to LPS (Figure 3d) support the notion that inflammation might play a role in Wwox related neuropathologies.

Recently, additional studies using other *Wwox* null mice models were reported, which support previous observations including: evidence of motor deficits, spontaneous epileptic seizures, and increased susceptibility to chemically-induced seizures in *Wwox-KO* mice [38]. Wwox deficient mice lagged behind WT littermates in reaching normal brain developmental stages, displaying microcephaly, incomplete separation of brain hemispheres, heterotopia, and defective cerebellar midline fusion as well as neuronal disorganization in histological examination. Hypomyelination with atrophy of the optic tract and cerebellar foliar white matter were also observed [38]. Interestingly, some of these observations agree with a recent report describing neuronal disorganization with defective architecture of the granular and molecular cell layers of cerebral cortex in brain histology samples from a *WWOX* null human fetus [39].

### 3.4. WWOX Association with Autism Spectrum Disorder, Intellectual Disability, and ADHD

ASD refers to a broad range of highly complex and heterogeneous neuropsychiatric conditions characterized by challenges with social skills, difficulty in communication and interaction with others, restricted interests, and repetitive behaviors. ADS individuals often suffer from other co-occurring conditions such as intellectual disability (ID), epilepsy, sleep disorders, motor deficits, and attention-deficit hyperactivity disorder (ADHD) [40]. Many of these ASD co-morbidities overlap with clinical features displayed by individuals affected with the above described WWOX related CNS disorders.

Until recently, recurrent chromosomal CNVs were recognized as the primary form of inherited risk variation for ASD. Association between the chr16q23.1 region containing *WWOX* and ASD can be first found in datasets of inherited CNVs in simplex families from the Simons Simplex Collection (SSC) [41], as in other large datasets focusing on CNVs in patients with intellectual and developmental disabilities [42]. More recently, using the Autism Genetic Resource Exchange (AGRE) and SSC datasets, Leppa et al. identified CNVs specifically spanning the *WWOX* locus in affected children from 12 families with multiple individuals with ASD out of a total of 3565 families (i.e., 0.34%), in comparison to only one unaffected sibling out of 2633 families (i.e., 0.04%, *p* = 0.01, odds ratio (OR) = 8.8). Of note, as per the Database of Genomic Variants (DGV), the overall frequency of >100 kb CNVs affecting *WWOX* was found to be 26 out of 27,263 individuals (i.e., 0.10%), which is significantly lower than the frequency observed with ASD individuals from multiplex families. Thus, the authors proposed that *WWOX* qualifies as a low-penetrance ASD associated locus [6]. Additional case reports and population-based studies support the association of *WWOX* point mutations and deletions with ID and ASD [40,43,44,45,46]. Figure 8 shows the distribution of *WWOX* locus CNVs associated with ASD cases (*n* = 49) obtained from the AutDB database (http://www.mindspec.org/autdb.html). The CNVs are mostly intragenic; however, additional larger variants with one of the breakpoints within the genomic region spanned by this gene are also observed. A summary of *WWOX* CNVs in ASD, ID, and developmental delay cases, chromosomal coordinates of the abnormalities, plus other relevant information is provided in Appendix A. Although the number of reports linking *WWOX* CNVs with ASD is limited, it is known that it would require very large population samples to reach significance for inherited low-penetrant ASD risk variants [47]. Nevertheless, according to the Simons Foundation Autism Research Initiative (SFARI) database, *WWOX* is classified as a Category 2 gene (Strong Candidate ASD gene), based on current data (https://gene.sfari.org).

Interestingly, WWOX also appears to be associated with ADHD. In a large meta-analysis of seven genome-wide linkage scans in search of loci associated with ADHD, it was determined that the chr16q23.1-qter region, precisely the genomic region containing *WWOX*, had a very highly significant risk association with ADHD [48]. Furthermore, Harich et al., using data from CNV studies from 6176 ADHD individuals and using multiple additional bioinformatic approaches, identified *WWOX* as one of a relatively small set of 26 candidate ADHD genes [49].

Germline CNVs affecting multiple loci contribute to human genetic and phenotypic diversity as well as play important roles in susceptibility to a broad spectrum of genetic and developmental disorders [50,51]. *WWOX* is indeed a hotspot for the spontaneous occurrence of germline polymorphic copy number variants in the human population. As per analysis of the database of human structural variation (dbVar) in healthy individuals from NCBI, it can be observed that the distribution of mostly intragenic germline *WWOX* CNVs in the normal population is significantly enriched, overlapping a clear hotspot within intron 5 of *WWOX* located at the 3′ edge of the core of chromosomal fragile site FRA16D (Figure 9) [52].

Importantly, *WWOX* has been recently identified as one of a very limited number of ‘long neural genes’ harboring recurrent DNA double strand breaks and as a consequence highly prone to undergoing genomic rearrangements in primary neural stem/progenitor cells. These genes susceptible to recurrent genomic break clusters are usually implicated both in neuropsychiatric disorders and in cancer. It is hypothesized that this breakage prone characteristic is related to critical processes likely associated with the specialized behavior of brain cells [53,54]. Thus, *WWOX* intrinsic and evolutionary conserved genomic fragility might represent a necessary feature of this gene as part of physiologic genomic rearrangements found in brain cells and proposed to play a role in the generation of neuronal diversity via somatic genomic mosaicism [55]. These observations emphasize the notion that *WWOX,* because containing FRA16D, is a hotspot for common germline CNVs and it is precisely this intrinsic fragility that is the cause for the frequent association of *WWOX* CNVs with neurological and developmental disorders as described here, i.e., WOREE, ASD, ID, and possibly other conditions, such as ADHD.

### 3.5. WWOX as a Risk Locus for Alzheimer’s Disease

Alzheimer’s disease (AD) is the most common cause of dementia (60–70%). It is estimated that over 44 million individuals worldwide and 5.8 million in the United States are living with AD, which is the sixth cause of death in the USA [56]. AD is a complex neurodegenerative disorder characterized by selective memory impairment as the most common and earliest clinical manifestation. In patients with the typical form of the illness, issues with other cognitive domains, including language deficits, visuospatial abnormalities, or decision-making functions, often occur later in the course of the disease.

In 2004, Sze et al. reported a potential association between WWOX and AD observing reduced WWOX protein levels in hippocampal neurons from AD patients compared to normal age-matched controls [57]. These authors also described that WWOX binds tau, suggesting a potential association with AD progression and proposed that loss of WWOX function modulates tau hyper-phosphorylation, activates the Aβ peptide aggregation cascade, and the generation of Aβ and amyloid fibrils [58].

The vast majority of AD cases develop in individuals 65 years or older; this is described as late-onset AD (LOAD). The main risk factors for LOAD are old age, family history of AD, and specific genetics factors [56]. In recent years, various genes have been identified and regarded as established risk loci for LOAD [59]. Recently, the International Genomics of Alzheimer’s Project (IGAP) performed an extensive GWAS meta-analysis comprehending more than 94,000 individuals (35,274 clinical and autopsy documented AD cases and 59,163 controls). This study confirmed 20 previously identified LOAD genome-wide risk loci and importantly led to the identification of 5 new additional risk genes: *IQCK*, *ACE*, *ADAM10*, *ADAMTS1*, and *WWOX*. Of relevance to this review, SNP variant rs62039712 (NC_000016.9: g.79355857G>A) mapping to the *WWOX* locus reached genome-wide significance (*p* = 3.7 × 10^−8^) with an OR of 1.16 [10]. The identification of *WWOX* as a novel AD risk gene is further supported by a recently published meta-analysis with the goal of identifying novel AD risk loci in African American individuals [60]. The authors identified several novel loci of genome-wide significance for individuals of African American ancestry. Interestingly, of the 25 previously identified AD loci in non-Hispanic White individuals [10], *WWOX* was found among only 7 loci that were implicated at a nominal significance level in African American individuals, suggesting that this gene may participate in cellular pathways associated with AD common to both ancestries [60].

Pathway analyses of common and rare variants AD gene risk subsets identified lipid metabolism among the major cellular biofunction clusters [10]. The most significant gene ontology (GO) pathways identified include: Protein–lipid complex assembly, reverse cholesterol transport, protein–lipid subunit organization, and plasma lipoprotein particle assembly. Further, the strong association of these lipid related pathways remained even after the removal of genes in the APOE related region [10]. These observations are in agreement with the functional themes reported on a separate AD GWAS meta-analysis which concluded that four main GO biofunctions were significantly associated with AD risk: Protein lipid complex; APP catabolic process; High-density lipoprotein (HDL) particle, Protein lipid complex assembly and with lower significance additional lipid-related themes such as Low-density lipoprotein (LDL) particle, Protein lipid complex organization and Chylomicron [61]. Importantly, *WWOX* is indeed associated with biofunctions related to lipid metabolism, homeostasis and transport. Genome-wide association studies led to the identification of a significant association between low levels of high-density lipoprotein cholesterol (HDL-C) and *WWOX* both in dyslipidemic families and when comparing low HDL-C cases and controls [62]. Other genome-wide studies confirmed the association between *WWOX*, HDL-C, and triglyceride levels [63,64,65]. Furthermore, disruption of *Wwox* using KO mice and liver targeted ablation leads to altered HDL and lipoprotein metabolism, suggesting that this protein plays a relevant role in cellular lipid homeostasis [66].

Activation of immune response was also identified as a significantly GO pathway associated to AD risk genes [10]. As described in previous sections, a direct association between WWOX and the NF-kB signaling pathway has been shown, and this protein was reported to be involved in various inflammatory conditions [24,25,26,27]. Interestingly, as earlier mentioned, *WWOX* expression is modulated by LPS in microglial cells (Figure 3d). Additionally, brain samples from *Wwox-KO* mice display evidence of activated microglia and astrogliosis [34]. Thus, we propose that WWOX not only belongs to the cluster of AD risk genes related to lipid metabolism and transport but also to biofunctions related to activation of the immune response. Further studies are warranted to gain mechanistic insight into the intriguing association of WWOX, lipids, neuroinflammation, and AD.

The IGAP study also investigated the association of specific traits and human behaviors that may be relevant to AD pathology. Among these traits, a significant negative correlation between AD risk and educational attainment as determined by multiple measures was identified, for example years of schooling, college completion and cognitive scores all behave as protective factors against AD (FDR *P* < 0.05) [10]. Interestingly, in recent large-scale GWAS of approximately 1.1 million individuals and as per using the MTAG method, various SNPs identified *WWOX* among several genome-wide significant loci associated with measures of educational attainment, including rs9937449 at a *p* value = 3 × 10^−13^ for years of education and rs7500549 with a *p* value = 6 × 10^−10^ for mathematical ability (highest math class taken) [67]. The association between a protective AD trait and *WWOX* adds an additional intriguing point of convergence linking this gene with cognitive skills.

### 3.6. WWOX, Multiple Sclerosis, and Myelination

The spectrum of neurodegenerative phenotypes associated with WWOX was recently further expanded with studies performed in individuals affected with multiple sclerosis (MS). MS is a chronic autoimmune disease of the CNS characterized by multifocal regions of demyelination, oligodendrocyte death, neurodegeneration, and varying degree of neurological dysfunction [68]. The first evidence of potential *WWOX* implication in MS was shown in an extensive GWAS performed with 14,498 MS subjects and 24,091 healthy controls to identify risk variants. The variants identified in the discovery phase were further validated in an independent cohort of 14,802 MS subjects and 26,703 healthy controls. In these 80,094 individuals, the rs12149527 (g.79076699C>A) intronic variant of *WWOX* was identified as one of 48 new susceptibility loci with genome-wide significance (*p* = 3.3 × 10^−11^) and an OR of 1.08 [69]. More recently, Matsushita et al. conducted a GWAS using cortical thickness as the outcome variable and pathway analyses using protein interaction networks; *WWOX* was found significantly associated with cortical thinning in 9 different brain regions in one of the MS patient cohorts [70]. In a targeted study of low-frequency variants in MS of Italian families, followed by studies in a cohort of 120 unrelated MS patients, statistically significant differences were observed in the comparison with control subjects for rs7201683. This SNP causes a p.Leu216Val missense variation in WWOX [71]. Recently, an additional variant mapping to the *WWOX* locus has been identified in another large-scale population-based study of 47,429 MS patients and 68,374 control subjects. The novel rs12925972 (g.79077400T>C) *WWOX* variant reached a genome-wide significance of *p* = 1 × 10^−19^ with an OR of 1.107 [72].

The myelin sheath is critical for the insulation of axons and transmission of the action potential, and it is produced by oligodendrocytes in CNS and Schwann cells in the PNS. Jakel et al., using single-nucleus RNA sequencing on white matter areas of post-mortem brains from MS patients with progressive disease vs. controls, observed differences in mature oligodendrocyte sub-clusters thus indicating possible different functional states in oligodendrocytes from MS lesions. In differential gene expression analyses, *WWOX* was detected as significantly downregulated in chronic active lesions of MS patients, data further validated by ISH [11]. This observation is interesting, as oligodendrocyte death due to autoimmune attack, and demyelination as a consequence, are the major pathological features of MS. Notably, in two recent studies in rodent models, an association between WWOX loss of function and aberrant myelination has been reported. In *Wwox-KO* mice, hypomyelination in several regions of the brain and peripheral nerves was observed [38]. While in the *Wwox* null *lde/lde* rat severe demyelination and a reduced number of mature oligodendrocytes were also described affecting the cerebral cortex [16]. Additionally, as mentioned earlier, MRIs of multiple WOREE patients displayed evidence of poor myelination as well. Recently, and of potential relevance to myelination processes, the intronic *WWOX* variant rs10514437 neared genome-wide significance (*p* = 1.56 × 10^−8^) for white matter infant brain volume in a GWAS conducted to identify common variants associated with infant brain volumes (*n* = 561 infants) [73].

Myelin is highly enriched in lipids (80%), including mainly glycosphingolipids, sphingomyelin, and cholesterol. Myelin biogenesis is a complex process in which protein and lipid trafficking, exocytosis, endocytosis, and endosomal recycling play critical roles [74,75]. Two mechanistic themes potentially link WWOX with myelogenesis. First, as mentioned in the AD section, there is strong evidence for WWOX to play a role in cellular lipid homeostasis [66]. Second, we have shown that WWOX protein is mostly found in the perinuclear cellular compartment significantly overlapping with the Golgi region [2,76]. Importantly, recently we demonstrated that WWOX directly interacts with multiple proteins involved in protein trafficking, endosome, and lysosomes networks, including SEC23IP, SCAMP3, VOPP1, and SIMPLE [76,77]. Notably, mutations affecting this latter protein, SIMPLE (small integral membrane protein of lysosome/late endosome), the earliest described WWOX interactor [76], were shown to cause the dominant demyelinating neuropathy CMTC1 [75,78].

The sum of the described observations suggests that a mechanistic association between WWOX and myelination likely exists and indeed points to *WWOX* as a strong candidate risk gene for involvement in MS pathogenesis.

## 4. Conclusions

The cerebellar cortex is the CNS structure displaying the highest WWOX expression levels as demonstrated by bulk tissue expression analyses and single-cell sequencing studies. The cerebellum receives information from different sensory systems, the spinal cord, and other regions of the brain. It coordinates almost all voluntary motor movements such as posture, balance, and speech, resulting in smooth and balanced muscular activity as well as eye movement and vision [79,80]. The observed higher WWOX expression in cerebellum and in specific cell types, such as basket cells and granule cells, is undoubtedly related to the observation that cerebellar ataxia is a syndromic condition associated with WWOX loss of function both in humans and rodent models.

Wwox expression is also specifically seen in the entorhinal cortex. This CNS region relays information between the neocortex and the hippocampus, and is responsible for learning and memory functions. Notably, severe abnormalities in the entorhinal cortex have been associated with Alzheimer’s disease, temporal lobe epilepsy, and cognitive impairment in humans [81,82,83], all conditions linked to WWOX. Similarly, the amygdala, where WWOX is also singularly expressed, is responsible for the perception of emotions as well as the controlling of aggression and its dysfunction is commonly associated with complex disorders such as autism and social incompetence, among other neuropsychiatric conditions [84,85]. Additionally, the amygdala is also affected early in neurodegenerative diseases such as Alzheimer’s disease [86].

Overall, rodent studies, so far, have established the importance of WWOX in normal CNS development and function, and its association with specific phenotypes. Wwox null rodent models reproduced many of the clinical features observed in WOREE patients; however, given the very limited lifespan of *Wwox-KO* mice, new conditional targeted models and models with longer lifespan are needed to provide more valuable mechanistic insights. For instance, targeted CNS *Wwox* ablation using promoters driving Cre recombinase to specific mouse brain regions such as the cerebellum, cortex, hippocampus, or specific cell types, such as excitatory neurons or microglia, will be highly informative. In addition, by means of CRISPR-Cas9 approaches, the generation of knock-in *Wwox* mutant mice emulating mutations found in humans would also provide significant pathobiological insight (studies ongoing in our laboratory).

Although CNS pathologies with demonstrable germline WWOX loss of function are rare, they provide strong evidence indicating a critical role for this gene in normal CNS development and function. Importantly, as discussed in this review, other far more common neuropathologies have also been linked to WWOX, such as Alzheimer’s disease, autism, and multiple sclerosis. In the mechanistic realm, information gained from diverse studies provided several hints associating WWOX with specific cellular pathways of much relevance in the various described neurogenerative disorders that deserve further detailed exploration, including lipid homeostasis, neuroinflammation, and proteolipid trafficking. Thus, further studies are necessary to gain insight on the role that WWOX plays in normal CNS development, function, and the broad array of pathologies so far linked to this multifaceted protein.

## Figures and Tables

**Figure 1 ijms-21-08922-f001:**
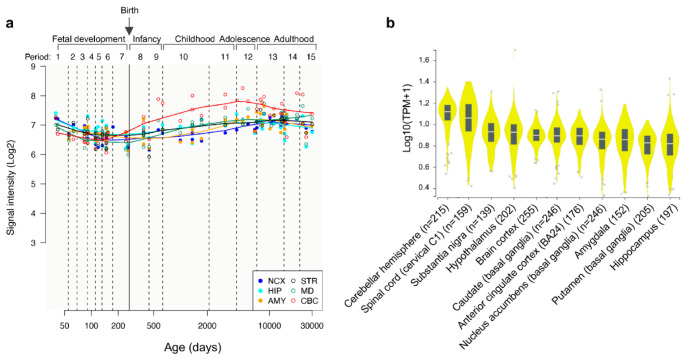
Temporospatial *WWOX* expression in human CNS tissues. (**a**) Trajectory plot showing the expression of *WWOX* during fetal development (period 1–7), infancy (period 8–9), childhood (period 10–11), adolescence (period 12), and adulthood (period 13–15), in individual CNS areas: neocortex (NCX), hippocampus (HIP), amygdala region (AMY), striatum (STR), mediodorsal nucleus of the thalamus (MD), and cerebellar cortex (CBC). Reprinted and adapted from Human Brain Transcriptome dataset (https://hbatlas.org/) [17]. (**b**) Violin plots showing the expression of *WWOX* in different brain regions as per the Genotype-Tissue Expression dataset (GTEx, Analysis Release V8, dbGaP Accession phs000424.v8.p2). Horizontal line in black box indicates the median value of *WWOX* expression in respective tissues; the number of samples for each tissue is indicated in the X-axis legend. TPM: Transcripts per Million. Reprinted and adapted from the RNA-seq GTEx database (https://gtexportal.org) [18].

**Figure 2 ijms-21-08922-f002:**
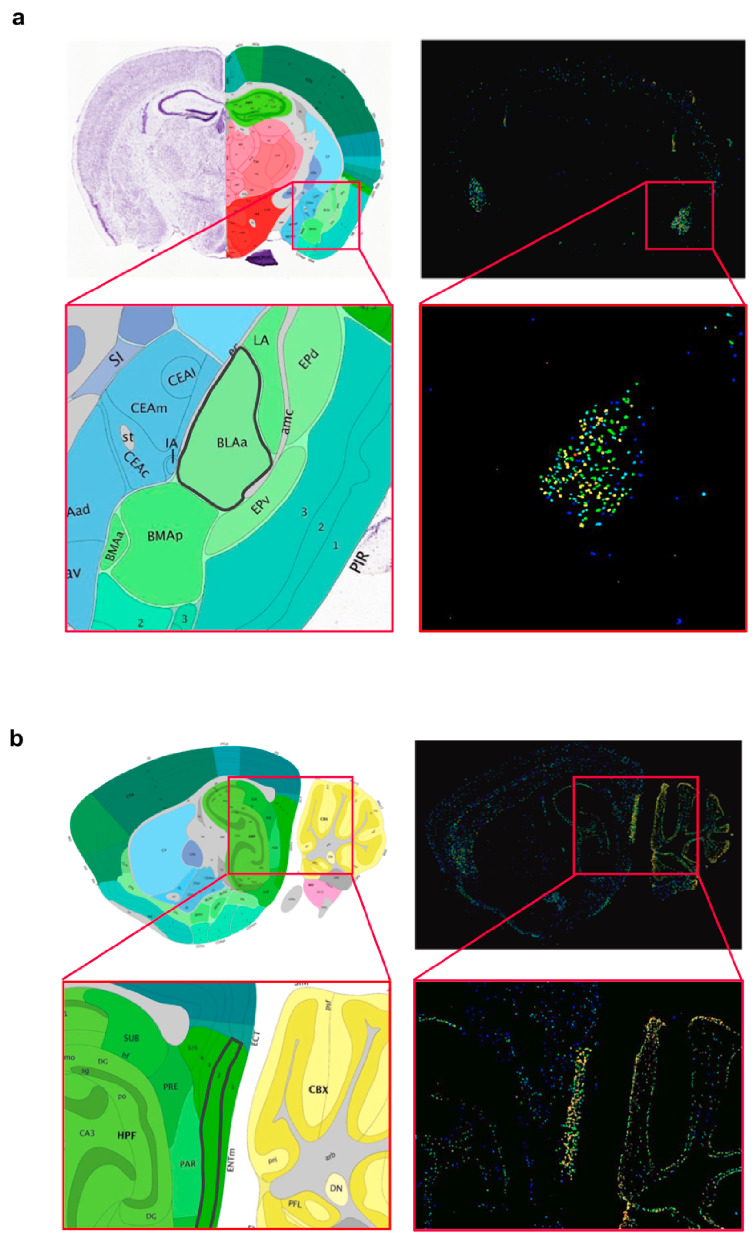
Localized *Wwox* expression in basolateral amygdala and medial entorhinal cortex of mouse brain. (**a**) Coronal brain section showing *Wwox* mRNA in-situ hybridization signals clearly delineating the anterior (BLAa) region of the basolateral amigdalar nucleus in the cortical subplate. The inset shows the zoomed image of the BLAa region, highlighted with a black border in the annotated panel (left). (**b**) Sagittal section showing in situ hybridization signals, specifically lighting-up layer 2 of the medial entorhinal cortex (ENTm2, medial part, dorsal zone, layer 2). The inset shows the zoomed image of the ENTm2 region, highlighted with a black border in the annotated panel (left). It can also be observed that *Wwox* expression clearly delineates specific layers of the cerebellar cortex. Brain tissue sections from 56-day old, C57Bl/6J male mouse. All images were obtained from the Allen Mouse Brain Atlas—Allen Institute (https://mouse.brain-map.org/) [19].

**Figure 3 ijms-21-08922-f003:**
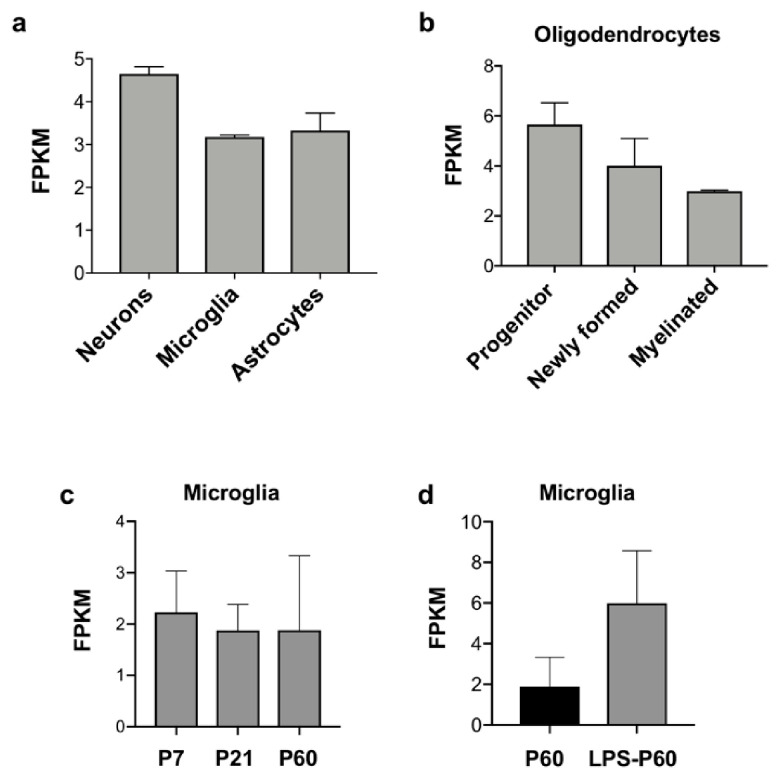
*Wwox* expression in mouse brain cell types. Bar graphs representing *Wwox* cell-type-specific expression levels, obtained from a mouse brain RNA-seq database (www.BrainRNAseq.org) [22,23] in (**a**) neurons, microglia, astrocytes, and (**b**) oligodendrocytes. Within the oligodendrocyte population, the highest *Wwox* expression levels are seen in progenitor oligodendrocytes, followed by newly formed and myelinated oligodendrocytes. (**c**) Uniform *Wwox* expression is observed at P7, P21, and P60 in microglial cells. (**d**) Wwox expression is upregulated upon LPS treatment. Y-axis represents Wwox expression as fragments per kilobase of transcript per million mapped reads (FPKM), error bars represent ± SD.

**Figure 4 ijms-21-08922-f004:**
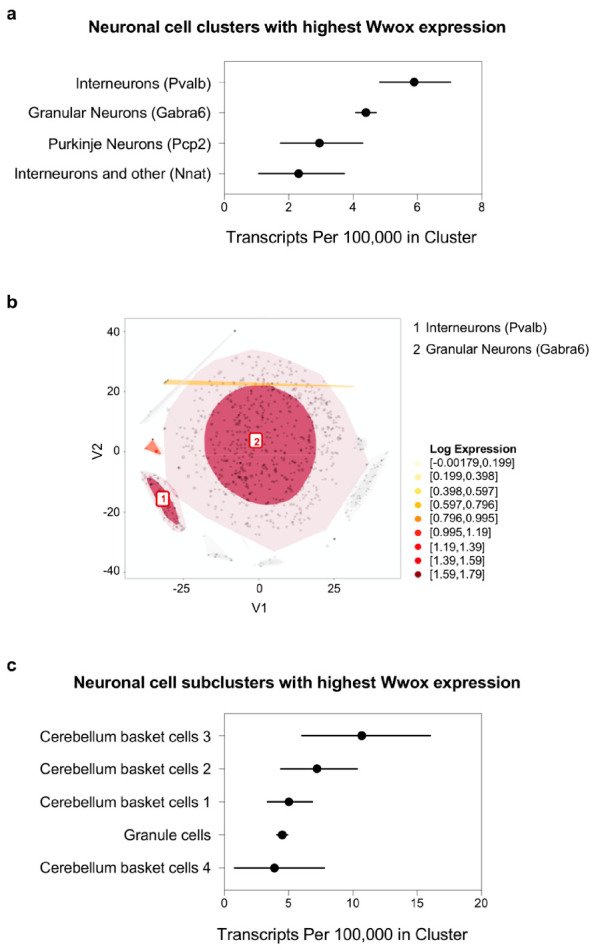
*Wwox* expression at single-cell level in mouse cerebellum neurons. (**a**) Graph representing neuronal cell clusters. Pvalb+ interneurons and Gabra6+ granule cells express the most *Wwox* transcripts in mouse cerebellum. Markers specific to each cell cluster are shown in parenthesis. (**b**) t-SNE plot highlights the top two clusters described in (**a**) as the darkest colored regions among cerebellum cells. (**c**) Graph representing specific neuronal subclusters expressing the most *Wwox* transcripts. GABAergic basket cells (i.e., interneurons) and granular neurons are the top cell types. The confidence intervals in graphs (**a**,**c**) reflect statistical sampling noise calculated from the binomial distribution and reflecting the total number of unique molecular identifier (UMIs) ascertained by cluster rather than cell-to-cell heterogeneity within a cluster. Data was obtained from the mouse brain single-cell RNA sequencing (scRNA-seq) DropViz database (http://dropviz.org/) [28].

**Figure 5 ijms-21-08922-f005:**
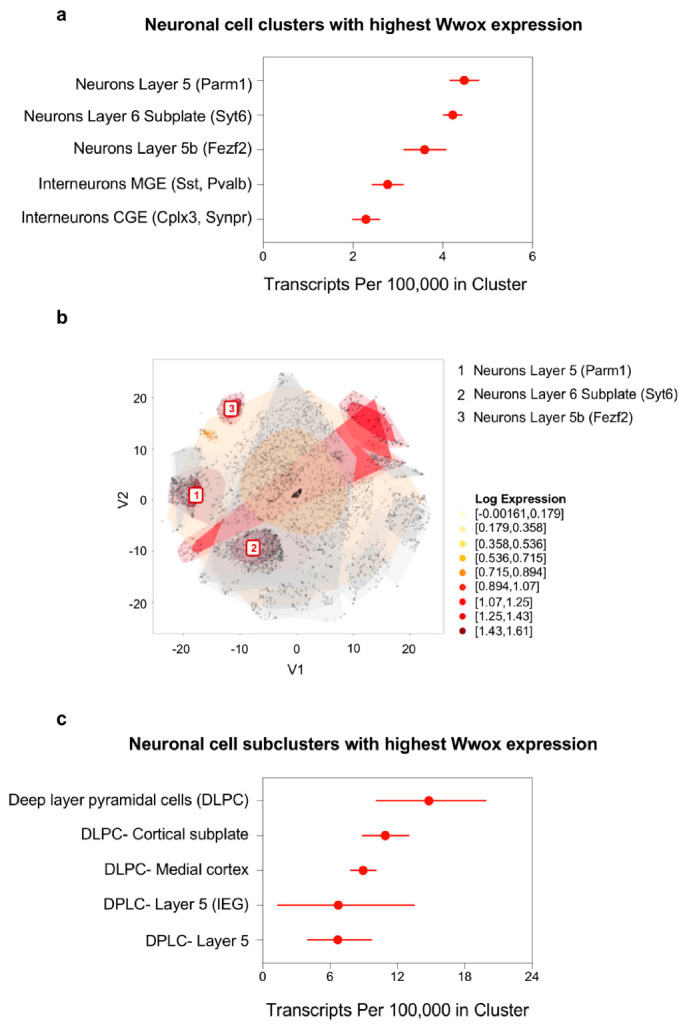
*Wwox* expression at single-cell level in mouse frontal cortex neurons. (**a**) Graph representing the neuronal cell clusters expressing the highest number of *Wwox* transcripts in mouse frontal cortex. Markers specific to each cell cluster are shown in parenthesis. Markers Parm1, Syt6 and Fezf2 identify the top three clusters. (**b**) t-SNE plot shows these three clusters as the darkest colored regions. (**c**) Graph representing the frontal cortex subclusters expressing the most *Wwox* transcripts, specifically deep-layer pyramidal neurons. The confidence intervals in graphs (**a**,**c**) reflect statistical sampling noise calculated from the binomial distribution and show the total number of UMIs ascertained by cluster. Data was obtained from the mouse brain scRNA-seq DropViz database (http://dropviz.org/) [28].

**Figure 6 ijms-21-08922-f006:**
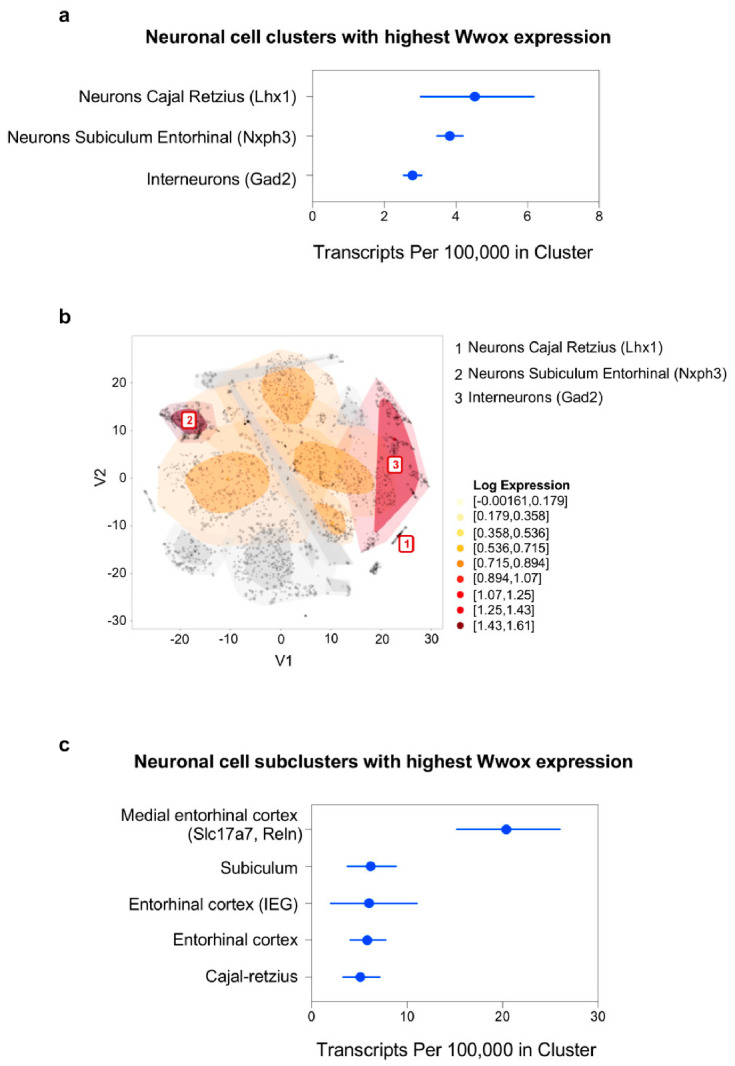
*Wwox* expression at single-cell level in mouse hippocampal neurons. (**a**) Graph representing the top three neuronal cell clusters expressing the highest number of *Wwox* transcripts identified by markers Lhx1, Nxph3 and Gad2 in mouse hippocampus. (**b**) These three clusters are shown in the t-SNE plot as the darkest colored regions. (**c**) Graph representing the hippocampal subclusters shows that neurons from the medial entorhinal cortex (markers Slc17a7 and Reln) express the most *Wwox* transcripts. The confidence intervals in graphs (**a**,**c**) reflect statistical sampling noise calculated from the binomial distribution and show the total number of UMIs ascertained by cluster. Data was obtained from the mouse brain scRNA-seq DropViz database (http://dropviz.org/) [28].

**Figure 7 ijms-21-08922-f007:**
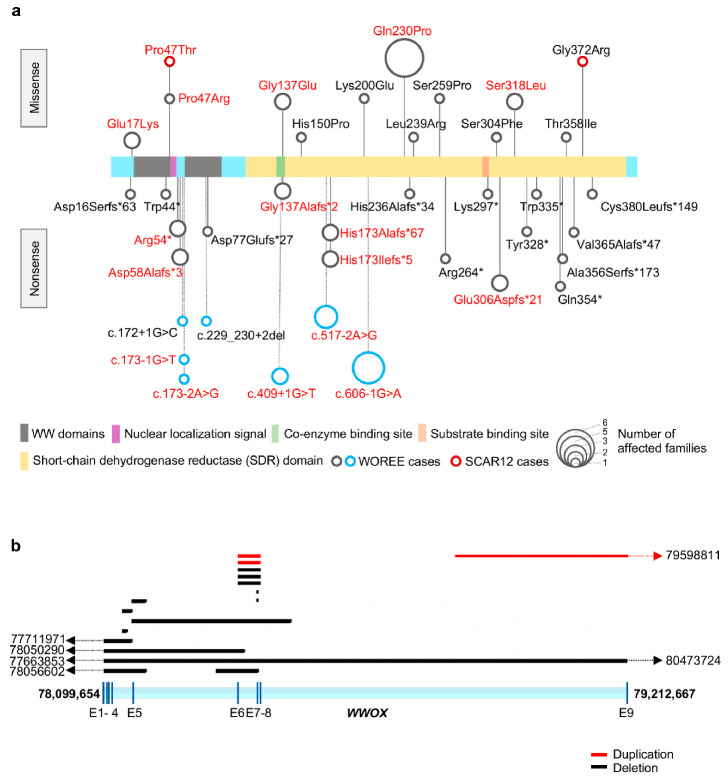
WWOX germline pathogenic variants in SCAR12 and *WWOX*-related epileptic encephalopathy (WOREE). (**a**) Missense, nonsense, and splice-site/intronic variants affecting WWOX protein in SCAR12 (red colored circles) and WOREE cases (gray colored circles for amino acid alterations and blue colored circles for splice-site/intronic variants). Size of each circle corresponds to the frequency of occurrence of the specific variant in single or multiple families, as noted. Mutation hotspots with more than one mutation at the same amino acid site or variants identified in more than one family are noted in red text. (**b**) Mapping of germline CNVs, duplications shown in red and deletions in black, of the *WWOX* locus in WOREE cases. Numbers next to dotted lines with arrowhead indicate coordinates of chromosomal breakpoints beyond the WWOX locus (human genome assembly GRCh38/hg38).

**Figure 8 ijms-21-08922-f008:**
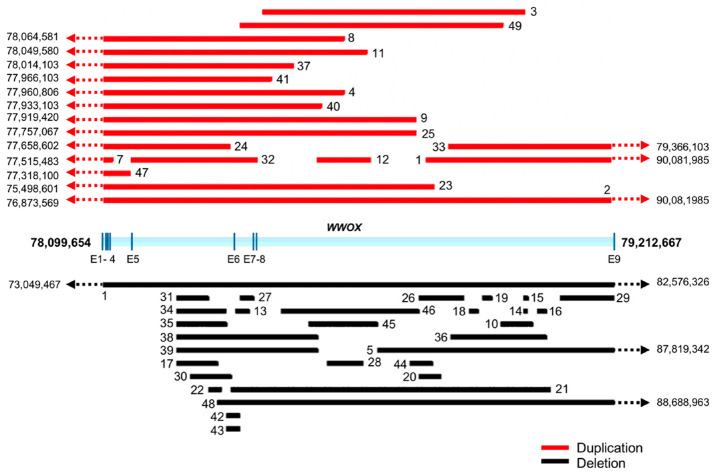
WWOX germline copy number variants (CNVs) in autism spectrum disorder (ASD). Distribution of *WWOX* locus CNVs associated with ASD and ID cases obtained from the AutDB database (http://www.mindspec.org/autdb.html). CNVs are mostly intragenic; however, additional larger variants with only one of the breakpoints within the genomic region spanned by this gene are also observed. Numbers next to dotted lines with arrowhead indicate coordinates of chromosomal breakpoints beyond the *WWOX* locus (human genome assembly GRCh38/hg38). Numbers 1–49 shown next to CNV bars correspond to case numbers described in Appendix A.

**Figure 9 ijms-21-08922-f009:**
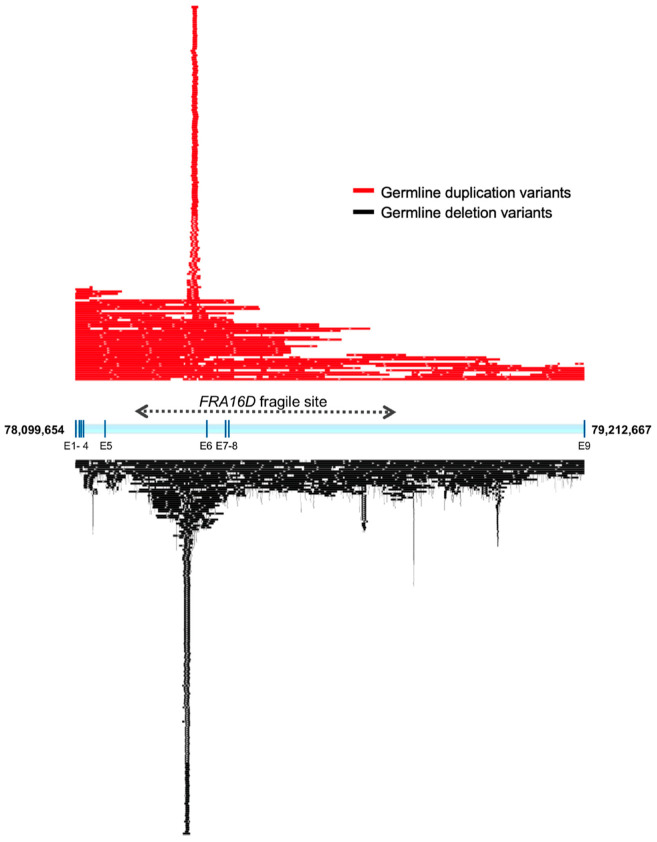
*WWOX* is a hotspot for germline CNV polymorphisms in healthy humans. Mapping of non-redundant *WWOX* intragenic germline duplications (red) and deletions (black) variants in normal human population. Larger CNVs spanning beyond *WWOX* with one of the breakpoints within the gene are also shown. As can be observed there is a significant accumulation of germline CNVs clustering in a specific hotspot within intron 5. This hotspot overlaps the 5′ prime edge of the core of *FRA16D* (dotted line).

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
