# Peer review of "WWOX Loss of Function in Neurodevelopmental and Neurodegenerative Disorders"

_ijms, 2020, doi:10.3390/ijms21238922_

Round 1

Reviewer 1 Report

The reviewed manuscript is a very insightful and detailed compendium of current knowledge on WWOX gene relevance for the functioning of the CNS.

The discussed topic is important and the publication perfectly describes what we know so far on the role of WWOX in brain pathology.

The Authors present an in-depth analysis of the available data on the expression of WWOX mRNA and protein in various parts of the nervous system, as well as in the cells of various types. They also discuss in detail the reported mutations and CNVs of the gene.

The data is presented in a legible way, the pictures have comprehensive captions.

The work is prepared very carefully in terms of editing. The only typo I could spot was a line 169 – “cluster” should be “clusters”.

Author Response

We thank the three reviewers for their effort in reviewing our manuscript and the positive comments received, such as Reviewer 1: “very insightful and detailed compendium of current knowledge on WWOX gene relevance for the functioning of the CNS”, that the topic “is important”. Reviewer 2: “The manuscript contains interesting remarks, is well written and organized”, and Reviewer 3: “The results are well presented, relevant, and encompassing pertinent knowledge on the topic”.

Point by point response to reviewer comments: 

Reviewer #1

Q- The work is prepared very carefully in terms of editing. The only typo I could spot was line 169 – “cluster” should be “clusters”.

A- The typographical error has been corrected.

Reviewer 2 Report

In this review, the authors describe the role of WWOX gene association with multiple central nervous system (CNS). Interestantly, is association of WWOX gene as a risk gene for common neurodegenerative conditions such as Alzheimer's disease (AD) and multiple sclerosis (MS). They evidenced that the gene expression of WWOX is comparatively higher in the human cerebellar cortex than in other CNS structures. Moreover, they showed that higher Wwox expression in interneurons and granule cells from cerebellum points to a direct link to the described cerebellar ataxia in cases of WWOX loss of function.

The manuscript contains interesting remarks, is well written and organized. The quality of pictures are very good and exaustive.

The conclusions are presented in an appropriate manner and are strongly supported. Finally, the article is written in standard and clear English.

Author Response

We thank the three reviewers for their effort in reviewing our manuscript and the positive comments received, such as Reviewer 1: “very insightful and detailed compendium of current knowledge on WWOX gene relevance for the functioning of the CNS”, that the topic “is important”. Reviewer 2: “The manuscript contains interesting remarks, is well written and organized”, and Reviewer 3: “The results are well presented, relevant, and encompassing pertinent knowledge on the topic”.

Reviewer #2

No corrections necessary

Reviewer 3 Report

The paper by Aldaz and Hussain (2020) reviews current knowledge on mutations in WWOX gene in relation to neurodevelopmental and neurodegenerative disorders. Known as a putative tumour suppressor, WWOX shows incredible pleiotropy with regards to neurological disorders, which makes this a valuable summary of investigations into this topic.

The article presents a balanced and comprehensive overview of neurological disorders related to WWOX loss of function. The account of data from publicly available databases provides a complete view on expression of WWOX in specific regions, layers and cell types of the brain. This is neatly linked to neurodevelopmental and neurodegenerative diseases and the brain regions and pathways affected in their development.

Minor improvement aspects (in bold letters)

The results are well presented, relevant, and encompassing pertinent knowledge on the topic from the last two decades, with mostly recent primary research. The figures are well designed and clear, however, in Figure 3d no indication of statistical significance is present, although it is mentioned in the text.

While the paper is a valuable compendium of knowledge on WWOX expression and loss of function in neurological disorders, it offers little critique of the experimental results cited. The conclusions lack a suggestion of a path forward or more specific experiments which could shed the light on this topic. However, synthesis and connection of the expression results from different databases with the brain regions involved in the disorders discussed is well outlined.

This paper is a well-designed, informative review with relevant data. Along with certain language and stylistic errors, a more clearly defined take away message and a suggestion of a path forward would be recommended.

Congratulations on your fine work

Author Response

We thank the three reviewers for their effort in reviewing our manuscript and the positive comments received, such as Reviewer 1: “very insightful and detailed compendium of current knowledge on WWOX gene relevance for the functioning of the CNS”, that the topic “is important”. Reviewer 2: “The manuscript contains interesting remarks, is well written and organized”, and Reviewer 3: “The results are well presented, relevant, and encompassing pertinent knowledge on the topic”.

Point by point response to reviewer comments: 

Reviewer #3

 Minor improvement aspects (in bold letters)

Q- The results are well presented, relevant, and encompassing pertinent knowledge on the topic from the last two decades, with the most recent primary research. The figures are well designed and clear; however, in Figure 3d, no indication of statistical significance is present, although it is mentioned in the text.

A-The necessary correction has been done.

Q- While the paper is a valuable compendium of knowledge on WWOX expression and loss of function in neurological disorders, it offers a little critique of the experimental results cited. The conclusions lack a suggestion of a path forward or more specific experiments that could shed light on this topic. However, the synthesis and connection of the expression result from different databases with the brain regions involved in the disorders discussed is well outlined.

This paper is a well-designed, informative review with relevant data. Along with certain language and stylistic errors, a more clearly defined take away message and a suggestion of a path forward would be recommended.

A- As suggested by the Reviewer, we have now included some specific statements in the Conclusions section of potential experiments that we are planning to conduct (some actually ongoing and well advanced) that we feel will provide a better mechanistic understanding of the role of WWOX in specific CNS regions and cell types and will allow furthering the narrow focus on key questions. The Conclusions section of the manuscript has been modified accordingly see lanes 588-597.